# Impact of the COVID-19 Pandemic on Ambulatory Care Antibiotic Use in Hungary: A Population-Based Observational Study

**DOI:** 10.3390/antibiotics12060970

**Published:** 2023-05-27

**Authors:** Helga Hambalek, Mária Matuz, Roxána Ruzsa, Zsófia Engi, Ádám Visnyovszki, Erika Papfalvi, Edit Hajdú, Péter Doró, Réka Viola, Gyöngyvér Soós, Dezső Csupor, Ria Benko

**Affiliations:** 1Institute of Clinical Pharmacy, Faculty of Pharmacy, University of Szeged, 6725 Szeged, Hungary; ruzsa.roxana@szte.hu (R.R.); zsofia.engi@szte.hu (Z.E.); papfalvi.erika@med.u-szeged.hu (E.P.); doro.peter@szte.hu (P.D.); viola.reka@szte.hu (R.V.); soosgyongyver@szte.hu (G.S.); csupor.dezso@szte.hu (D.C.); benko.ria@med.u-szeged.hu (R.B.); 2Central Pharmacy Department, Albert Szent-Györgyi Health Center, University of Szeged, 6725 Szeged, Hungary; 3Department of Internal Medicine Infectiology Unit, Albert Szent-Györgyi Health Centre, University of Szeged, Állomás Street 1–3, 6725 Szeged, Hungary; visnyovszki.adam@szte.hu (Á.V.); horvathne.hajdu.edit@med.u-szeged.hu (E.H.); 4Institute for Translational Medicine, Medical School, University of Pécs, 7624 Pécs, Hungary; 5Emergency Department, Albert Szent-Györgyi Health Centre, University of Szeged, Semmelweis Street 6, 6725 Szeged, Hungary

**Keywords:** antibiotics, COVID-19, pandemic, restrictions, outpatient, DDD, antibiotic utilization, trend, pattern

## Abstract

The COVID-19 pandemic and related restrictions have potentially impacted the use of antibiotics. We aimed to analyze the use of systemic antibiotics (J01) in ambulatory care in Hungary during two pandemic years, to compare it with pre-COVID levels (January 2015–December 2019), and to describe trends based on monthly utilization. Our main findings were that during the studied COVID-19 pandemic period, compared to the pre-COVID level, an impressive 23.22% decrease in the use of systemic antibiotics was detected in ambulatory care. A significant reduction was shown in the use of several antibacterial subgroups, such as beta-lactam antibacterials, penicillins (J01C, −26.3%), and quinolones (J01M, −36.5%). The trends of antibiotic use moved in parallel with the introduction or revoking of restriction measures with a nadir in May 2020, which corresponded to a 55.46% decrease in use compared to the previous (pre-COVID) year’s monthly means. In general, the systemic antibiotic use (J01) was lower compared to the pre-COVID periods’ monthly means in almost every studied pandemic month, except for three months from September to November in 2021. The seasonal variation of antibiotic use also diminished. Active agent level analysis revealed an excessive use of azithromycin, even after evidence of ineffectiveness for COVID-19 emerged.

## 1. Introduction

The coronavirus disease 2019 (COVID-19) pandemic and related restrictive measures, such as school closures and stay-at-home orders, had an impact on the epidemiology of infectious diseases and substantially affected healthcare and medication use. During the pandemic period, enacted rules on physical distancing may have reduced the transmission of viral and bacterial pathogens and, consequently, the number of patients (for example, in the case of influenza or pneumococci infections).

Several studies were published that analyzed antibiotic use during the first year of the COVID-19 pandemic [1]. Research from the USA estimated the monthly number of patients who dispensed antibiotic prescriptions from retail pharmacies from January 2017 through May 2020 [2]. The ambulatory use of antibiotics in the first year of the pandemic was also examined in their respective countries by Canadian, Scottish, and Portuguese researchers [3,4,5,6]. A Belgian study aimed to assess the impact of COVID-19 on GPs’ antibiotic prescriptions during out-of-hours consultations [7]. However, research is scarce that summarizes the changes in antibiotic use during the two pandemic years. Only Dutch colleagues monitored antibiotic prescription trends during both years of the COVID-19 pandemic in general practice [8].

We aimed to assess the antibiotic utilization trends/patterns in outpatient care in Hungary during the two pandemic years, and to compare it to pre-COVID levels.

## 2. Results

During the whole study period, 288 million DDD systemic antibiotics were used in the outpatient sector in Hungary. The national use of antibiotics was 12.10 DDD per 1000 inhabitants per day in the pre-COVID period, while during the COVID-19 period, this decreased to 9.29 DDD per 1000 inhabitants per day (a 23.22% decrease).

Table 1 summarizes the absolute and relative use of antibiotics. We observed a notable decline in the use of most antibiotic subgroups, except for tetracyclines. The reduction in the use of penicillins (J01C), from 4.15 to 3.06 DDD per 1000 inhabitants per day (26.3%) and in the use of quinolones from 2.22 to 1.41 DDD per 1000 inhabitants per day (36.5%), was the largest.

The pattern of antibacterial use changed slightly between the two periods. The relative use of macrolides increased from 16.53% to 20.34% of the total systemic ambulatory antibiotic use. Moreover, the relative use of tetracyclines (J01A) increased from 6.36% to 8.35%. However, we observed a slight decline in the relative use of quinolones (J01M) (from 18.35% to 15.15%) (Table 1).

The top ten list of antibacterials in the two main periods are displayed in Table 2. No significant changes were observed: the same active agents were on the top of the list before and during the pandemic (pre-COVID and COVID-19 period), and co-amoxiclav led both with a substantial relative share. In the ranking, only some changes were detected: cefuroxime dropped from second to third place, while azithromycin became the second most used antibacterial with an over 15% relative share. Clindamycin moved from the ninth position to the sixth. Regarding quinolones, two agents, levofloxacin and ciprofloxacin, were in the top list. 

In Figure 1, we display the monthly use of systemic antibiotics (J01) during the COVID-19 subperiods (from period 0 to period 5) and the corresponding monthly values (minimum, maximum, mean) of the pre-COVID period (2015–2019).

In general, the systemic antibiotic use (J01) was lower compared to the pre-COVID periods’ mean in almost every pandemic month, except for three months from September to November 2021 (period 4).

The main differences in the scale of systemic antibiotic use (J01) between the pandemic and the corresponding pre-COVID monthly means were detected in period 1 and 3. For example, during pre-COVID May, the mean was 10.62 DDD per 1000 inhabitants per day, while the antibiotic use in May 2020 (period 1) was 4.73 DDD per 1000 inhabitants per day, a drop of 55.46%.

In period 1, after the start of the state of emergency, we observed a significant decrease in antibiotic use, with a nadir in May 2020. However, the use of antibiotics slightly increased from June 2020 (period 2), and the expected seasonal winter peak did not occur. In period 3, the highest antibiotic use was detected in March 2021, but this was still significantly lower than the pre-COVID monthly mean for this month.

Figure 2 shows the monthly use of different antibiotic subgroups (J01C, J01D, J01F, J01M) during the COVID-19 period and the corresponding monthly mean values of the pre-COVID period.

Regarding the two beta-lactam subgroups, both penicillin and cephalosporin use followed the trends of systemic antibiotic use. In April 2021, the utilization of co-amoxiclav (J01CR02) was reduced by half. This decreased use persisted until September 2021, when the previous use resumed. Macrolide use during the COVID-19 periods had different trends. It first peaked in March 2021, and from November 2021, it consequently and significantly exceeded the pre-COVID monthly means. The highest peak in the use of macrolides was in November 2021 (4.71 DDD per 1000 inhabitants per day). In the use of azithromycin, we also observed a peak in March 2021, and then a significant increase in September 2021. This higher utilization was retained until the end of the research period.

The subgroup of quinolones was the only antibacterial subgroup where the monthly scale of antibiotic use never reached the pre-COVID period’s monthly means.

## 3. Discussion

To our knowledge, this study was the first to analyze ambulatory care antibiotic use in a Central European country during the COVID-19 pandemic, and one of the drug utilization studies that covered two years of the pandemic (March 2020–February 2022). Considering this fact, it should be noted that any comparison should be made carefully due to the different study periods and outcome measures used.

Our main findings were that during the COVID-19 period, an impressive 23.22% decrease in the total use of systemic antibiotics in ambulatory care was detected compared to the pre-COVID period.

The use of systemic antibiotics in ambulatory care also decreased in other countries during the pandemic. Pandemic-related changes in antibiotic use in 2020 were analyzed by Portuguese colleagues. They found a substantial decrease in ambulatory care antibiotic prescription, reaching 9.65, 9.62, and 7.04 DDD per 1000 inhabitants per day in 2018, 2019, and 2020, respectively. This decrease corresponded to 26.94% and was in line with the magnitude of the decrease observed in Hungary. As Hungary has continuously been among the EU countries with a lower antibacterial consumption rate in the community in the pre-COVID years according to ECDC reports, it is especially important to identify the leading reason for the further reduction in antibacterial consumption in this sector in Hungary during the pandemic [9]. It may be partly explained by the collaterally reduced incidence of other respiratory tract infections (RTIs) as a consequence of restrictive/preventive pandemic-related measures such as social distancing, isolation, the routine use of facial masks, or mandatory home confinement periods. A previous Hungarian study has demonstrated that there was significant seasonal variation in the use of systemic antibiotics in Hungary, with significantly higher antibiotic use during the winter months [10]. The seasonal variation of antibacterial use in Hungary was proven to be one of the highest in Europe and indicates the prescription of antibiotics for self-limiting viral respiratory tract infections (RTIs) [11,12]. The lack of seasonal increase in the winter months during the pandemic indirectly explained the lower prescription rate for viral RTIs. More difficult access to health care and the consequent lower number of General Practitioners’ (GP) consultation rates could also have contributed to decreased antibiotic use [6,8]. In Hungary, the annual average consultation rates in the pre-COVID period (2015–2019) were 66,625,096 (face-to-face visits in doctors’ office or at patient’s home), which dropped to an average of 61,080,883 during the two pandemic years (2020, 2021), which is an 8.3% decrease. However, in the second year of the pandemic (2021), it was possible for doctors to register telemedicine consultations (mainly telephone calls), and if we incorporate those into the statistics, we end up with 68,147,204 consultations, which is a 2.3% increase in the consultation rate compared to the pre-COVID period [13].

Analyzing monthly values, the use of systemic antibiotics (J01) decreased by more than 45% in Portugal during the first pandemic months (April, May 2020) compared to the similar period from the two previous years, which is similar to the decline of 55.46% in the Hungarian data from May 2020 compared to the pre-COVID monthly means [6]. In Canada, the highest decrease in national antibiotic prescriptions dispensed per 1000 inhabitants was detected in May 2020, corresponding to a 39.56% drop in comparison to May 2019 [4]. In a Belgian study focusing only on out-of-hours services, it was found that the number of antibiotic prescriptions per weekend per 100,000 inhabitants ranged from minus 11.5 to minus 42.9% immediately following the implementation of the lockdown [7].

Dutch colleagues aimed to monitor antibiotic prescription trends during the COVID-19 pandemic in general practice. They analyzed antibiotic prescription rates per 100,000 inhabitants per week. They assessed changes in antibiotic prescribing during different pandemic phases, which follows the Hungarian chronology of defined periods well (see Methods). In comparison to the pre-COVID levels, the mean number of antibiotic prescriptions in the Netherlands decreased by 17.3% in Phase 1 and by 22.74% in Phase 3 [8]. Quantification is not possible due to the different outcome measures, but this is similar to the pattern of trends in Hungary (a substantial decrease during period 1 and period 3).

A substantial decline in the use of many antibacterial subgroups was detected. The most frequently used J01C subgroup, the penicillin combinations including beta-lactamase inhibitors (J01CR), showed a nearly 20% drop in use in the COVID-19 period compared to the pre-COVID means. In Canada, the mean monthly number of prescriptions for penicillin with beta-lactamase inhibitors exhibited a 21.85% decrease during the pandemic [3]. The largest difference was in prescriptions of co-amoxiclav in quarter 4 (October to December 2020): the observed monthly mean was 40% less than expected [14].

In the use of second-generation cephalosporins (J01DC), we observed a decrease from 1.70 to 1.08 DDD per 1000 inhabitants per day, and a decrease in proportional use from 14.05% to 11.62%. In the Canadian study, they found that second- or third-generation cephalosporin use decreased by 52.7% between the pre-COVID and the COVID-19 period, whereas in the US, the decrease in the use of cephalosporins was 33%, the third largest decrease from antibiotic classes (corresponding to 1.1 million fewer patients) [2,3].

Fluoroquinolones were consumed less in the COVID-19 period (a 36.5% decrease in use compared to the pre-COVID level). In the US, fluoroquinolone use decreased by 18% during the pandemic, while in Canada, the mean monthly number of second- and third-generation fluoroquinolone prescriptions decreased by 25.4% and 58.4%, respectively [2,3].

In the use of macrolides and lincosamides (J01F), we detected two peaks: first in March 2021 and then in November 2021, when it significantly exceeded the mean pre-COVID monthly value. Azithromycin became the second most used antibacterial in Hungary during the pandemic. Azithromycin has been proposed for the treatment for COVID-19 on the basis of its immunomodulatory and antiviral actions, which may have boosted the use of macrolides during the pandemic [15,16,17]. The first concerns were published in August 2020 [18], and then, in March 2021, azithromycin was proven to be ineffective against COVID-19, so it was only recommended for bacterial superinfection, as are other antibiotics [19]. In contrast to the Hungarian trends, in Canada, a marked reduction was observed (a 65% decrease) in macrolide prescribing during the COVID-19 period (from March to December 2020) compared with the pre-COVID period (from January 2017 to February 2020) [3]. The first peak in the use of azithromycin in Hungary was in March 2021. Unfortunately, we observed an increase in the use of azithromycin in September 2021, and then the highest peak in the use of macrolides was in November 2021. On this basis, we can conclude that the evidence on the ineffectiveness of azithromycin against COVID-19 in Hungary was not adopted well and in a timely manner by prescribers.

Several discussed studies assessed the indications of antibiotics. The Canadian colleagues analyzed the prescribing rates of antibiotics for different indications including urinary tract infections (UTIs) [4]. Indication-linked antibiotic use was not available in our dataset, but commonly used antibiotics for UTIs (i.e., fosfomycin, nitrofurantoin) had a decreased use in Hungary, similar to the lower Canadian prescription rates for UTIs.

### Strengths and Limitations

One of the main strengths of this research is the longitudinal nature of the dataset and the entire coverage of the two “main” pandemic years (2020, 2021). Secondly, we used a population-level dataset with excellent coverage (both in terms of inhabitants and systemic antibiotics). We have to acknowledge the limitations of this research. First, the results of this study may not be generalizable to other settings or locales because of the great diversity in how restrictions were applied during the COVID-19 pandemic. Secondly, we did not have access to either weekly level antibiotic use data or indication-linked data, which precluded the application of interrupted time serial analysis and the analysis of infection types. Thirdly, we used an aggregated dataset of prescribed and reimbursed systemic antibiotics, which precluded the quantification of the number of prescriptions per inhabitants. Lastly, we did not have data on individual prescribers, so we could not compare prescribing trends and pattern changes on an individual level.

## 4. Materials and Methods

The data were obtained from the public database of the National Health Insurance Fund (Hungarian acronym NEAK) [20]. NEAK is the sole insurance fund in Hungary with almost 100% population coverage. The NEAK database contains each antibiotic prescription redeemed in the Hungarian community pharmacies.

As antibiotics are prescription-only medicines in Hungary, and with the exception of a few products, they are reimbursed medicines, the database provides 95% drug coverage.

Package-level data of redeemed prescriptions were retrieved for monthly intervals. This study included antibiotics for systemic use, i.e., J01 products, as defined by the WHO’s Anatomical Therapeutic Chemical (ATC) Classification Index (version 2022) [21]. The data were stratified by antibiotic subgroup. The package-level data were converted, and finally expressed as defined daily doses (DDD) per 1000 inhabitants per day (DID). We defined two main periods: the pre-COVID period (from January 2015 to December 2019) and the COVID-19 pandemic period (from January and February 2020, when no cases were reported officially and from March 2020 to February 2022, when the pandemic was declared). For the pre-COVID period, we calculated the 5-year mean of systemic antibiotic use for each month and compared it to the corresponding months of the pandemic period. In addition, to be able to better analyze antibiotic use in relation to restriction measures, we further defined six different subperiods (see below) during the COVID-19 period. Antibiotic utilization data were compared between the two main periods and then further analyzed during the six different COVID subperiods. Hungary initially declared a state of emergency on 11 March 2020. The Hungarian government made strict restrictions related to the COVID-19 pandemic. The restrictions changed periodically according to the actual epidemiological situation. We collected information on the Hungarian pandemic restrictions and stratified antibiotic use as follows: period 0: 2020 January and February, period 1: from March 2020 to May 2020, period 2: from June 2020 to October 2020, period 3: from November 2020 to April 2021, period 4: from May 2021 to October 2021, and period 5: November 2021 to March 2022. Details of the different restriction measures are detailed in Box 1 (Appendix A). In summary, in periods 1 and 3, we had strict lockdowns, with school closures, restrictions on gatherings, mask wearing in outside places, and social distancing of 1.5–2 m in every public place. In other periods, moderate restriction measures were in place, the curfew expired, and we had access to social, cultural, or sports programs, with certain limitations. Vaccination against COVID-19 became available during the third period [22]. After that, we could only enter public indoor events by proving proper vaccination status.

Box 1COVID-19 and restrictive measures timeline in Hungary [22,23,24,25].Period 1 (March 2020–May 2020): First lockdownThe first COVID-19 patient in Hungary was reported on 4 March 2020. Measures included social distancing (1.5–2 m in public places) and the closing of kindergartens, while schools and universities switched to online education. Restaurants were open until 3 pm, providing only delivery services. Shopping centers were closed, except for pharmacies, grocery stores, bakeries, and markets, with an isolated purchase period for elderly people. Sport and culture events were canceled. Weddings and funerals were limited to close relatives. Hotels, cinemas, museums, libraries, and sport facilities were also closed. There was a curfew order that meant people could only leave their homes for a legitimate reason, for example, in case of emergency or work duties. People had to wear face masks in indoor spaces.Period 2 (June 2020–October 2020):The intermediate phase during summer with fewer infections and fewer restrictions. Restaurants could open the terraces and, later, the inner spaces. Shops were open with longer opening hours. Sports and cultural events were organized with a 1.5 m protecting distance, and weddings and funerals were held with a maximum of 200 participants. Outdoor events could be organized with up to 500 participants. Curfew was suspended. However, wearing a mask was obligatory in indoor places. Schools maintained online education until the end of the school year.Period 3 (November 2020–April 2021): Second lockdownThe second and third wave of infections came with more strict lockdown measures, including social distancing. At first, high schools and universities switched to online education; after one month, this was extended to all educational institutions. Restaurants, hotels, cinemas, theaters, and sport facilities were closed. A general ban on events was in place, so cultural and sports events were canceled, while weddings and funerals could be organized with just 50 participants. Shops were first open until 7 pm, and later, all shops were closed, except for bakeries, grocery stores, and pharmacies. Curfew was in place between 8 p.m. and 5 a.m. Mask wearing was obligatory in outside places, too. Vaccination began in January 2021.Period 4 (May 2021–October 2021):The second intermediate phase had lighter restrictions again, except for basic guidance regarding hand hygiene and social distancing. Schools slowly returned to in-person education. Visiting hotels, cinemas, museums, and cultural and sports events was only permitted with a vaccination card. Weddings and funerals could be organized with maximum of 200 participants. Outdoor events could be organized again with up to 500 participants.Period 5 (November 2021–March 2022): Third lockdownThe fourth and fifth wave of infections passed with lighter measures, e.g., hand hygiene fostering measures and mandatory face masks in indoor places. In-person education returned to schools. Shops opened with regular opening hours. Events were permitted for people with a vaccination card, with 200 participants in inner spaces and a maximum of 500 participants outdoors.

## 5. Conclusions

This population-level study highlights the significant impact of the COVID-19 pandemic on the antibiotic use trends in ambulatory care for the first time in a Central European country. The COVID-19 pandemic-related restrictions resulted in a significant decrease in antibiotic use in the Hungarian ambulatory care sector. The pattern changes were less substantial, and revealed a decreased significance of fluoroquinolones and a widespread use of macrolides (e.g., azithromycin) during the pandemic, even after the proven ineffectiveness of the latter for COVID-19. Monthly changes of antibiotic use followed the introduced and revoked pandemic-related social restrictions well. Some of the used measures during the pandemic (e.g., compulsory facial mask wearing) should be considered by policy makers to control the spread of other infectious diseases (e.g., influenza epidemics) in the future, which could consequently reduce the related antibiotic utilization. In the future, health care professionals should uptake the changing recommendations of evidence-based medicine more rapidly. Another recommendation for practicing doctors is to integrate telemedicine into their daily work for diagnosing certain infectious diseases (e.g., for acute cystitis that does not require physical examination or compulsory laboratory tests). The long-lasting impact of the COVID-19 pandemic on antibiotic use should be investigated in further research.

## Figures and Tables

**Figure 1 antibiotics-12-00970-f001:**
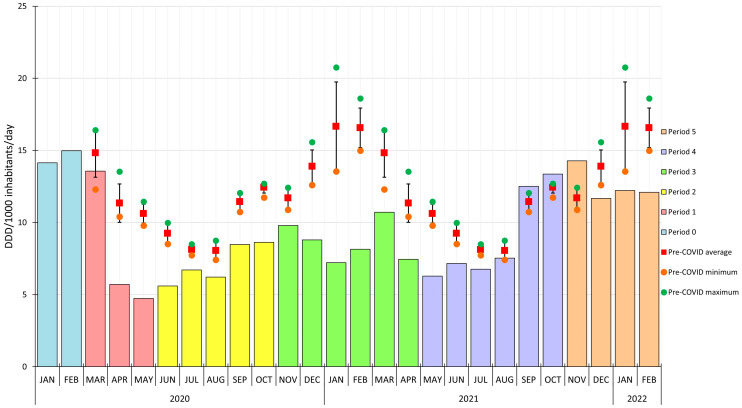
Red squares represent the mean monthly antibiotic use in the pre-COVID period (years 2015–2019). The orange circles represent the minimum, while the green circle represents the pre-COVID period’s maximum value. The whiskers represent the standard deviation. The colored columns represent the monthly use of antibacterials in the different COVID-19 periods.

**Figure 2 antibiotics-12-00970-f002:**
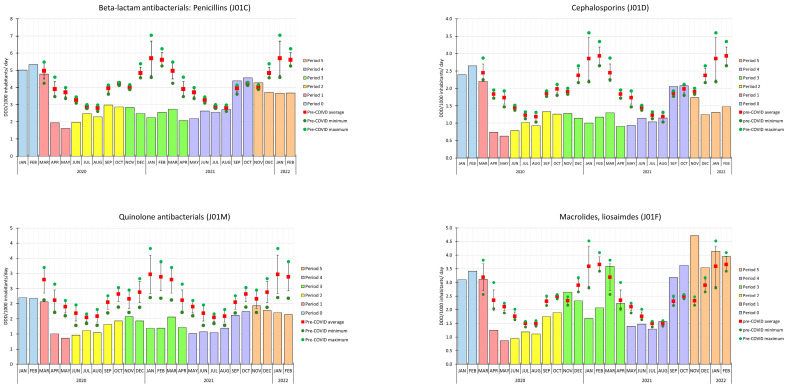
Red squares mean the mean monthly antibiotic use in the pre-COVID period (year 2015–2019). The orange circles represent the minimum, while the green circle represents the pre-COVID period’s maximum. The colored columns represent the use of antibacterials in the different COVID-19 periods.

**Table 1 antibiotics-12-00970-t001:** Means of systemic antibiotic use in the two main periods.

	Pre-COVID Period		COVID-19 Period	
	DID ^1^	%	DID ^1^	%
J01A Tetracyclines	0.77	6.34	0.78	8.35
J01CA Penicillins with extended spectrum	0.61	5.04	0.29	3.12
J01CE Beta-lactamase sensitive penicillins	0.17	1.41	0.06	0.65
J01CR Combinations of penicillins, incl. beta-lactamase inhibitors	3.37	27.85	2.71	29.17
J01C Beta-lactam antibacterials, penicillins	4.15	34.28	3.06	32.91
J01DC Second-generation cephalosporins	1.70	14.05	1.08	11.62
J01DD Third-generations cephalosporins	0.27	2.23	0.24	2.58
J01D Cephalosporins	1.97	16.25	1.32	14.24
J01E Sulfonamides and trimethoprim	0.43	3.57	0.36	3.88
J01FA Macrolides	2.00	16.53	1.89	20.34
J01FF Lincosamides	0.50	4.13	0.47	5.06
J01F Macrolides, lincosamides	2.49	20.59	2.35	25.34
J01M Quinolone antibacterials	2.22	18.35	1.41	15.15
J01X Other antibacterials	0.06	0.53	0.01	0.07
J01 Antibacterials	12.10	100.00	9.29	100.00

^1^ DID: DDD per 1000 inhabitants per day.

**Table 2 antibiotics-12-00970-t002:** Top list of antibacterial use.

	Pre-COVID Period					COVID-19 Period				
No.	ATC Code	Substance	DID ^1^	%	Cum% ^2^	ATC Code	Substance	DID ^1^	%	Cum% ^2^
1.	J01CR02	AMC ^3^	3.36	27.80	27.80	J01CR02	AMC ^3^	2.70	29.10	29.10
2.	J01DC02	cefuroxime	1.34	11.12	38.92	J01FA10	azithromycin	1.41	15.14	44.24
3.	J01FA10	azithromycin	1.22	10.05	48.97	J01DC02	cefuroxime	0.84	9.07	53.31
4.	J01MA12	levofloxacin	1.10	9.07	58.04	J01AA02	doxycycline	0.78	8.35	61.66
5.	J01AA02	doxycycline	0.77	6.34	64.38	J01MA12	levofloxacin	0.64	6.93	68.58
6.	J01FA09	clarithromycin	0.73	6.05	70.44	J01FF01	clindamycin	0.47	5.04	73.63
7.	J01MA02	ciprofloxacin	0.66	5.46	75.89	J01MA02	ciprofloxacin	0.46	5.00	78.62
8.	J01CA04	amoxicillin	0.61	5.01	80.91	J01FA09	clarithromycin	0.46	4.97	83.59
9.	J01FF01	clindamycin	0.50	4.09	85.00	J01EE01	SMX-TMP ^4^	0.36	3.88	87.47
10.	J01EE01	SMX-TMP ^4^	0.43	3.57	88.57	J01CA04	amoxicillin	0.29	3.08	90.55

^1^ DID: DDD per 1000 inhabitants per day; ^2^ cum%: cumulative percentage; ^3^ AMC: amoxicillin and clavulanic acid; ^4^ SMX-TMP: sulfamethoxazole and trimethoprim.

## Data Availability

Data are available from the corresponding author upon reasonable request.

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
