# Peer review of "Impact of the COVID-19 Pandemic on Ambulatory Care Antibiotic Use in Hungary: A Population-Based Observational Study"

_antibiotics, 2023, doi:10.3390/antibiotics12060970_

Round 1

Reviewer 1 Report

This study investigated a critical issue, which is the impact of COVID-19 on antibiotic use. However, the following issues need to be addressed before considering this manuscript for publication:

Discussion: Although the decreased rate of antibiotic use in this study was compared with the reported rate in the previous studies, the possible reasons or factors for this change in antibiotic use were compared between the current and previous studies. Specifically, how could the change in antibiotic use between the pre-COVID and COVID periods be used to improve antimicrobial utilization in the future and propose novel interventions for antimicrobial stewardship?

Supplementary Materials: Four supplementary figures have been included. However, these figures are the same as those in Figure 2 in the main manuscript. What is the purpose of including these figures twice?

Lines 37 – 38: This sentence used the abbreviation “e.g.” incorrectly. It is better to spell the word “for example” to make the sentence clearer.

Box 1 – (1. Period (03.2020-05.2020): First lockdown): The wrong tense was used for the verb “cancell”.

The whole manuscript requires revision due to several grammatical mistakes, including missing commas, wrong verb tenses, and difficult-to-read sentence structures.

Reviewer 2 Report

Comments to the Authors,

Minor comments:

Line 22: or quinolones should be replaced with and quinolones.

Line 58: Tetracyclines should be replaced with tetracyclines.

Line 136 and 203: Portuguese collegues should be replaced with Portuguese colleagues.

Line 158: Please respect the space between words. "this is".

Questions:

Is it possible to sell antibiotics in pharmacies without a doctor's prescription? It is suggested that this be mentioned in the text.

Can the changes in antibiotic prescription be reported in two time periods in total between different cities? In case of a positive answer, which cities had the greatest decrease or increase in antibiotic consumption?

Are the statistics of prescriptions prescribed by the doctor similar in two time periods? If there are statistics, please mention them in the text, and if there are no statistics, it is suggested to refer to this point in the limitations section.

Reviewer 3 Report

Dear Authors,

This article contains significant and novel findings that are worth publication. However, minor corrections are required before considering this article for publication. Please refer to my comments in the attached file.

Kind regards,

Reviewer 4 Report

Dear Authors,

The article with the title: "Impact of the COVID-19 pandemic on ambulatory care antibiotic use in Hungary: a population based observational study" is an interesting overview of ambulatory care antibiotic consumption in Hungary during the covid-19 pandemic.

The presented research offers and interesting insight in antimicrobial consumption during pandemic since it includes both years of the most intense covid-19 pandemic and the results of antimicrobial consumption are displayed and analyzed month by month and compared with pre-covid consumption during the same months.

It also provides an information about an interesting dynamic of antimicrobial selection during the course of pandemic (e.g. the use of azithromycin, etc.), although in depth analysis is missing.

The ECDC annual report of Antimicrobial consumption for 2020 already shows the phenomenon of reduced consumption of antibiotics in some EU countries during the first year of covid-19 and among them Hungary as well. Seven countries (Estonia, Greece, Hungary, Italy, Latvia, Malta, Portugal) reported a decrease in the community, but unfortunately an increase in the hospital sector.

According to the ECDC reports, Hungary has also been among the EU counties with a lower antimicrobial consumption rate in community already in pre-covid era (2011-2020), which is very good health-care indicator and therefore it is especially important to find out what was the leading reason for the further reduction in antimicrobial consumption in this sector in Hungary during the pandemic.

I would encourage the authors to expand the research and to include in the data presentation and analysis monthly data on the number of health-care visits at the primary care/general practitioners during the pandemic and during the pre-covid era, as this would provide an important additional information which could influence the antimicrobial consumption rate.

In discussion the authors should discuss different reasons and factors influencing the lower antimicrobial consumption during pandemic more precisely. Many factors could influence antibiotic prescription: on one hand the covid-19 preventive measures have collaterally reduced the incidence of other respiratory tract infection, whereas on the other a reduced accessibility of the health system was (probably) seen as well, other factors?

The monthly analysis of azithromycin consumption is very interesting and is really a good indicator of not so good treatment trends during the pandemic which we have unfortunately observed in many countries.

The authors must improve the citation of the literature.

Best regards!

Reviewer 5 Report

The paper deals with the impact of the Pandemics on the outpatient antibiotic use in Hungary, for the first time. Topic is highly relevant, and the paper identifies the effects of pandemics on the changes in antibiotic use during 2 years of pandemics, analyzing specific time periods with respect of pandemic waves, lock downs and other control measures in Hungary.  In my opinion, article is suitable for acceptance in the present form due to the fact that:   -the study design is appropriate -all methods are clearly reported -well written and supported with adequate references -clearly addressed strength and limitations

Round 2

Reviewer 1 Report

The authors addressed my previous comments appropriately—no further comments.

Reviewer 4 Report

Dear Editor!

The authors improved the manuscript significantly. I recommend accepting the manuscript.

Best regards,

Nina Gorišek Miksić